# VIDEODIRECTORGPT: CONSISTENT MULTI-SCENE VIDEO GENERATION VIA LLM-GUIDED PLANNING

## ABSTRACT

Although recent text-to-video (T2V) generation methods have seen significant advancements, the majority of these works focus on producing short video clips of a single event with a single background (i.e., single-scene videos). Meanwhile, recent large language models (LLMs) have demonstrated their capability in generating layouts and programs to control downstream visual modules such as image generation models. This prompts an important question: *can we leverage the knowledge embedded in these LLMs for temporally consistent long video generation?* In this paper, we propose VIDEODIRECTORGPT, a novel framework for consistent multi-scene video generation that uses the knowledge of LLMs for video content planning and grounded video generation. Specifically, given a single text prompt, we first ask our video planner LLM (GPT-4) to expand it into a '*video plan*', which involves generating the scene descriptions, the entities with their respective layouts, the background for each scene, and consistency groupings of the entities and backgrounds. Next, guided by this output from the video planner, our video generator, named Layout2Vid, has explicit control over spatial layouts and can maintain temporal consistency of entities/backgrounds across multiple scenes, while being trained only with image-level annotations. Our experiments demonstrate that our proposed VIDEODIRECTORGPT framework substantially improves layout and movement control in both single- and multi-scene video generation and can generate multi-scene videos with visual consistency across scenes, while achieving competitive performance with SOTAs in open-domain single-scene text-to-video generation. We also demonstrate that our framework can dynamically control the strength for layout guidance and can also generate videos with user-provided images. We hope our framework can inspire future work on integrating the planning ability of LLMs into consistent long video generation.

## 1 INTRODUCTION

Text-to-video (T2V) generation has achieved rapid progress following the success of text-to-image (T2I) generation. Most works in T2V generation focus on producing short videos (e.g., 16 frames at 2fps) from the given text prompts (Wang et al., 2023b; He et al., 2022; Ho et al., 2022; Singer et al., 2023; Zhou et al., 2022). Recent works on long video generation (Blattmann et al., 2023; Yin et al., 2023; Villegas et al., 2023; He et al., 2023) aim at generating long videos of a few minutes with holistic visual consistency. Although these works could generate longer videos, the generated videos often display the continuation or repetitive patterns of a single action (e.g., driving a car) instead of transitions and dynamics of multiple changing actions/events (e.g., five steps about how to make caraway cakes). Meanwhile, large language models (LLMs) (Brown et al., 2020; OpenAI, 2023; Touvron et al., 2023a;b; Chowdhery et al., 2022) have demonstrated their capability in generating layouts and programs to control visual modules (Dídac et al., 2023; Gupta & Kembhavi, 2023), especially image generation models (Cho et al., 2023b; Feng et al., 2023). This raises an interesting question: *Can we leverage the knowledge embedded in these LLMs for planning consistent multi-scene video generation?*

In this work, we introduce VIDEODIRECTORGPT, a novel framework for consistent multi-scene video generation. As illustrated in Fig. 1, VIDEODIRECTORGPT decomposes the T2V generation task into two stages: **video planning** and **video generation**. For the first video planning stage (see Fig. 1 blue part), we employ an LLM to generate a *video plan*, which is an overall plot of the video

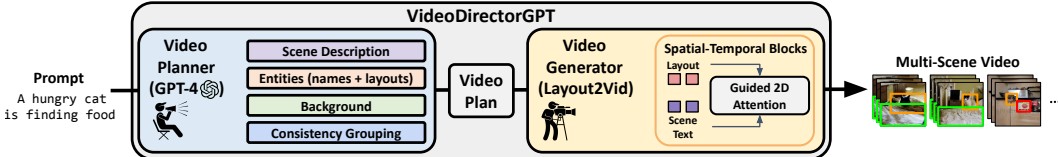

Figure 1: **Overall illustration of our VIDEODIRECTORGPT framework.** In the first stage, we employ **GPT-4 as a video planner** to craft a ***video plan***, which provides a multi-component script for videos with multiple scenes. In the second stage, we utilize **Layout2Vid**, a grounded video generation module, to render multi-scene videos with layout and consistency control based on the *video plan* generated in the first stage.

with multiple scenes, each consisting of a text description of the scene and entity names/layouts, and a background. It also consists of consistency groupings of specific entities/backgrounds that re-appear across scenes. For the second video generation stage (see Fig. 1 yellow part), we introduce Layout2Vid, a novel grounded video generation module that generates multi-scene videos from the *video plan*. Our framework provides the following strengths: (1) employing an LLM to write a *video plan* that guides the generation of videos with multiple scenes from a single text prompt, (2) layout control in video generation by only using image-level layout annotations, and (3) generation of visually consistent entities/backgrounds across multiple scenes.

To be specific, in the first stage, video planning (Sec. 3.1), we employ an LLM (e.g., GPT-4 (OpenAI, 2023)) as a video planner to generate a ***video plan***, a multi-component script of videos with multiple scenes to guide the downstream video synthesis process. Our *video plan* consists of four components: (1) multi-scene descriptions, (2) entities (names and their 2D bounding boxes), (3) background, and (4) consistency groupings (scene indices for each entity/background indicating where they should remain visually consistent). We generate the *video plan* in two steps by prompting an LLM with different in-context examples. In the first step, we expand a single text prompt into multi-step scene descriptions with an LLM, where each scene is described with a text description, a list of entities, and a background (see Fig. 2 blue part for details). We also prompt the LLM to generate additional information for each entity (e.g., color, attire, etc.), and group together entities across frames and scenes, which will help guide consistency during the video generation stage. In the second step, we expand the detailed layouts of each scene with an LLM by generating a list of bounding boxes of the entities per frame, given the list of entities and scene description. This overall '*video plan*' guides the downstream video generation module in the second stage (described next).

In the second stage, video generation (Sec. 3.2), we introduce Layout2Vid, a grounded video generation module to render videos based on the *video plan* generated by the LLM in the previous stage (see yellow part of Fig. 2). For the grounded video generation module, we build upon ModelScopeT2V (Wang et al., 2023b), an off-the-shelf text-to-video generation model, by freezing its original parameters and adding spatial control of entities through a small set of trainable parameters (13% of total parameters) in the gated-attention module (Li et al., 2023). This enables our Layout2Vid to be trained solely with layout-annotated images, thus bypassing the need for expensive training on annotated video datasets. To preserve the identity of entities appearing across different frames and scenes, we use shared representations for the entities within the same consistency group. We also propose to use a joint image+text embedding as entity grounding conditions which we find more effective than the existing text-only approaches (Li et al., 2023) in entity identity preservation (Appendix F). Overall, our Layout2Vid avoids expensive video-level training and also improves the object layout and movement control and cross-scene temporal consistency.

We conduct experiments on both single-scene and multi-scene video generation. For single-scene video generation, we evaluate layout control via VPEval Skill-based prompts (Cho et al., 2023b), assess object dynamics through ActionBench-Direction prompts adapted from ActionBench-SSV2 (Wang et al., 2023c), and examine open-domain video generation using the MSR-VTT dataset (Xu et al., 2016). For multi-scene video generation, we experiment with two types of input prompts: (1) a list of sentences describing events – ActivityNet Captions (Krishna et al., 2017) and Coref-SV prompts based on Pororo-SV (Li et al., 2019b), and (2) a single sentence from which models generate multi-scene videos – HiREST (Zala et al., 2023). Experiments show that our proposed VIDEODIRECTORGPT demonstrates better layout skills (object, count, spatial, scale) and object movement control (Sec. 5.1), capable of generating multi-scene videos with visual consis-

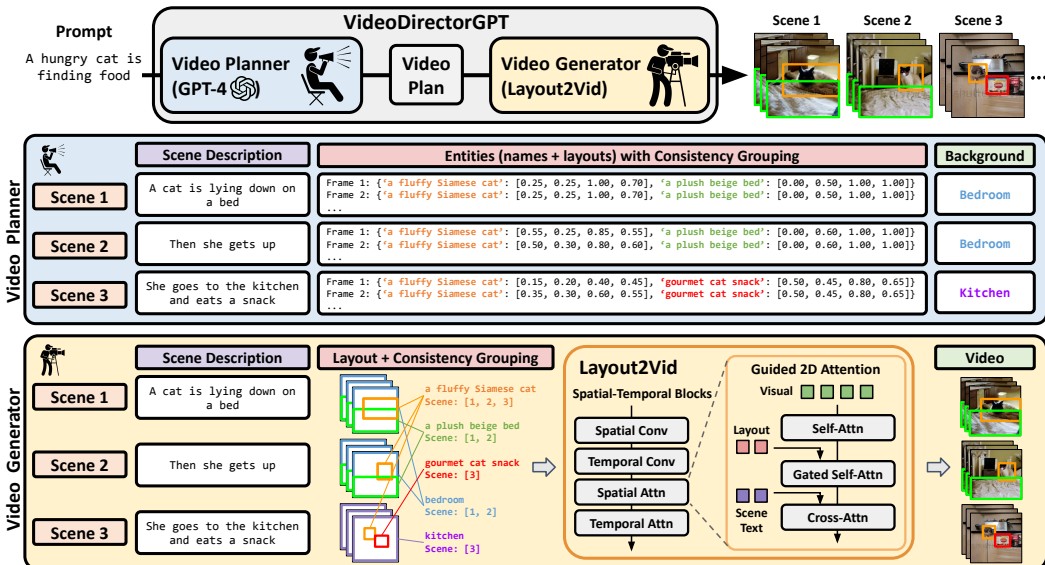

Figure 2: **Illustration of our two-stage framework for long, multi-scene video generation from text.** In the first stage, we employ the **LLM as a video planner** to craft a *video plan*, which provides an overarching plot for videos with multiple scenes, guiding the downstream video generation process (Sec. 3.1). The *video plan* consists of scene-level text descriptions, a list of the entities and background involved in each scene, frame-by-frame entity layouts (bounding boxes), and consistency groupings for entities and backgrounds. In the second stage, we utilize **Layout2Vid**, a grounded video generation module, to render videos based on the *video plan* generated in the first stage. This module uses the same image and text embeddings to represent identical entities and backgrounds from *video plan*, and allows for spatial control over entity layouts through the Guided 2D Attention in the spatial attention block (Sec. 3.2).

tency across different scenes (Sec. 5.2), and competitive with SOTAs on single-scene open-domain text-to-video generation (Sec. 5.1). We also demonstrate that our framework can dynamically control the strength for layout guidance and generate videos with user-provided images (Sec. 5.3).

Our main contributions can be summarized as follows:

- We propose a new T2V generation framework VIDEODIRECTORGPT with two stages: video content planning and grounded multi-scene video generation.
- We employ LLMs to generate a '*video plan*' which consists of detailed scene descriptions, entity layouts, and entity/background consistency groupings to guide downstream video generation (Sec. 3.1).
- We introduce Layout2Vid, a novel grounded video generation module, which brings together image/text-based layout control ability and entity-level temporal consistency (Sec. 3.2). Our Layout2Vid can be trained using only image-level layout annotations.
- We empirically demonstrate that our framework can accurately control object layouts and movements in single-scene videos (Sec. 5.1) and can generate temporally consistent multi-scene videos (Sec. 5.2). We also provide qualitative examples, ablation study of our design choices (Appendix F), and human evaluations (Sec. 5.4).

## 2 RELATED WORKS

**Text-to-video generation.** Training a text-to-video (T2V) generation model from scratch is computationally expensive. Recent work often leverages pre-trained text-to-image (T2I) generation models such as Stable Diffusion (Rombach et al., 2022) by fine-tuning them on text-video pairs (Wang et al., 2023b; Blattmann et al., 2023). While this warm-start strategy enables high-resolution video generation, it comes with the limitation of only being able to generate short video clips, as T2I models lack the ability to maintain consistency through long videos. On the other hand, recent works on long video generation (Blattmann et al., 2023; Yin et al., 2023; Villegas et al., 2023; He et al., 2023)

aim at generating long videos of a few minutes. However, the generated videos often display the continuation or repetitive patterns of a single action instead of transitions and dynamics of multiple changing actions/events. In contrast, our layout-guided T2V generation model, Layout2Vid, infuses layout control and multi-scene temporal consistency into a pretrained T2V generation model via data and parameter-efficient training, while preserving its original visual quality.

**Bridging text-to-image generation with layouts.** To achieve interpretable and controllable generation, a line of research decomposes the T2I generation task into two stages: text-to-layout generation, and layout-to-image generation. While early models train the layout generation module from scratch (Hong et al., 2018; Tan et al., 2019; Li et al., 2019a; Liang et al., 2022), recent methods employ pretrained LLMs to leverage their knowledge in generating image layouts from text (Cho et al., 2023b; Feng et al., 2023; Qu et al., 2023). To the best of our knowledge, our work is the first to utilize LLMs to generate structured video plans from text, enabling accurate and controllable long video generation. See Appendix A for addtional related works.

## 3 VIDEODIRECTORGPT

### 3.1 VIDEO PLANNING: GENERATING VIDEO PLANS WITH LLMS

**Video Plan.** As illustrated in the blue part of Fig. 2, GPT-4 (OpenAI, 2023) acts as a planner, providing a detailed *video plan* to guide the video generation. Our *video plan* has four components: (1) **multi-scene descriptions**: a sentence describing each scene, (2) **entities**: names and their 2D bounding boxes, (3) **background**: text description of the location of each scene, and (4) **consistency groupings**: scene indices for each entity/background indicating where they should remain visually consistent. The *video plan* is generated in two steps by prompting GPT-4 independently. See Appendix B for each step's GPT-4 prompt details.

**Video Planning Step 1: Generating multi-scene descriptions, entity names, and entity/background consistency groupings.** In the first step, we employ GPT-4 to expand a single text prompt into a multi-scene *video plan*. Next, we group entities and backgrounds that appear across different scenes using an exact match. For instance, if the 'chef' appears in scenes 1-4 and 'oven' only appears in scene 1, we form the entity consistency groupings as {chef:[1,2,3,4], oven:[1]}. In the subsequent video generation stage, we use the shared representations for the same entity/background consistency groups to ensure they maintain temporally consistent appearances (see Sec. 3.2 for details).

**Video Planning Step 2: Generating entity layouts for each scene.** In the second step, we expand the detailed layouts for each scene using GPT-4. We generate a list of bounding boxes for the entities in each frame based on the entities and the scene description. For each scene, we produce layouts for 8 frames, then linearly interpolate to gather information for denser frames (e.g., 16 frames).

### 3.2 VIDEO GENERATION: GENERATING VIDEOS FROM VIDEO PLANS WITH LAYOUT2VID

**Layout2Vid: Layout-guided T2V generation.** We implement Layout2Vid by integrating layout control capability into ModelScopeT2V (Wang et al., 2023b), a public T2V generation model based on Stable Diffusion (Rombach et al., 2022) (see Appendix C.1 for ModelScopeT2V details). The diffusion UNet in ModelScopeT2V consists of a series of spatio-temporal blocks, each containing four modules: spatial convolution, temporal convolution, spatial attention, and temporal attention. Compared with ModelScopeT2V, our Layout2Vid enables layout-guided video generation with explicit spatial control over a list of entities represented by their bounding boxes, as well as visual and text content. As illustrated in Fig. 3 (a), we build upon the 2D attention to create the Guided 2D Attention. As shown in Fig. 3 (b), the Guided 2D Attention takes two conditional inputs to modulate the visual latent representation: (a) layout tokens, conditioned with gated self-attention (Li et al., 2023), and (b) text tokens that describe the current scene, conditioned with cross-attention.

**Temporally consistent entity grounding with image+text embeddings.** While previous layout-guided text-to-image generation models commonly used only the CLIP text embedding for layout control (Li et al., 2023; Yang et al., 2023), we use the CLIP image embedding in addition to the CLIP text embedding for entity grounding. In our ablation studies (see Appendix F), we find that using both the image and text embeddings for grounding is more effective than text-only or image-only grounding. As depicted in Equation (1), the grounding token for the $i^{th}$ entity, $\boldsymbol{h}_i$, is a 2-layer

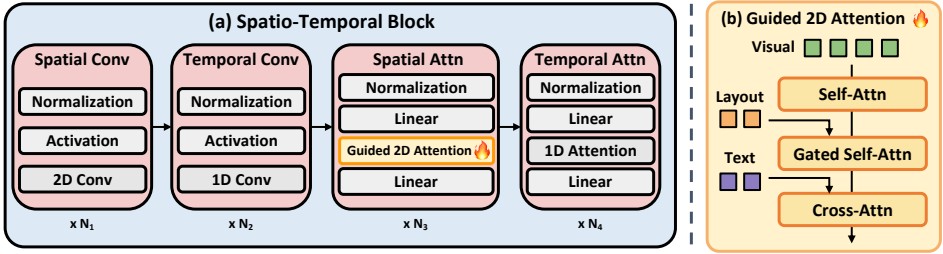

Figure 3: Overview of **(a) spatio-temporal blocks** within the diffusion UNet of our **Layout2Vid** and **(b) Guided 2D Attention** in the spatial attention module. (a) The spatio-temporal block comprises four modules: spatial convolution, temporal convolution, spatial attention, and temporal attention. We adopt settings from ModelScopeT2V, where (N1, N2, N3, N4) are set to (2, 4, 2, 2). In (b) Guided 2D Attention, we modulate the visual representation with layout tokens and text tokens. For efficient training, only the parameters of the Guided 2D Attention (indicated by the fire symbol, constituting 13% of total parameters) are trained using image-level annotations. The remaining modules in the spatio-temporal block are kept frozen.

MLP which fuses CLIP image embeddings $\boldsymbol{f}_{\text{img}}(e_i)$, CLIP text embeddings $\boldsymbol{f}_{\text{text}}(e_i)$, and Fourier features (Mildenhall et al., 2021) of the bounding box $\boldsymbol{l}_i = [x_0, y_0, x_1, y_1]$. We use learnable linear projection layers, denoted as $\boldsymbol{P}_{\text{img/text}}$, on the visual/text features, which we found helpful for faster convergence during training in our initial experiments.

$$\boldsymbol{h}_i = \text{MLP}(\boldsymbol{P}_{\text{img}}(\boldsymbol{f}_{\text{img}}(e_i)), \boldsymbol{P}_{\text{text}}(\boldsymbol{f}_{\text{text}}(e_i)), \text{Fourier}(\boldsymbol{l}_i)) \qquad (1)$$

Our image embedding $\boldsymbol{f}_{\text{img}}(e_i)$ can be obtained from either text descriptions or user-provided exemplar images. To obtain image embeddings from text (e.g., from the *video plan*), we employ Karlo (Lee et al., 2022), a public implementation of unCLIP Prior (Ramesh et al., 2022), which translates a CLIP text embedding into a CLIP image embedding. To obtain the image embedding from image exemplars, we can simply encode the images with the CLIP image encoder.

**Parameter and data-efficient training.** During training, we only update the parameters of the Guided 2D Attention (13% of total parameters) to inject layout guidance capabilities into the ModelScopeT2V backbone while preserving its original video generation capabilities. Such a training strategy allows us to efficiently train the model with only image-level layout annotations, while still equipped with multi-scene temporal consistency via shared entity grounding tokens. Training and inference details for our Layout2Vid are shown in Appendix C.2.

## 4 EXPERIMENTAL SETUP

**Evaluated models.** We primarily compare our VIDEODIRECTORGPT with ModelScopeT2V, and present comparisons with other T2V generation models (see Appendix D.1 for all baseline model details) on the datasets for which their papers have provided results. ModelScopeT2V serves as the closest baseline to our framework, given that our Layout2Vid utilizes its frozen weights and only trains a small set of new parameters to add spatial control and temporal consistency across scenes.

**Prompts for single-scene video generation.** For single-scene video generation, we conduct experiments with VPEval Skill-based prompts (Cho et al., 2023b), (which cover skills including object, count, spatial relations, and relative scale) to evaluate layout control, ActionBench-Direction prompts to assess object dynamics, and MSR-VTT captions to cover diverse open-domain scenes (Xu et al., 2016). Specifically, we prepare ActionBench-Direction prompts by sampling video captions from ActionBench-SSV2 (Wang et al., 2023c) and balancing the distribution of movement directions. See Appendix D.2 for details.

**Prompts for multi-scene video generation.** For multi-scene video generation, we experiment with two types of input prompts: (1) a list of sentences describing events – ActivityNet Captions (Krishna et al., 2017) and Coref-SV prompts based on Pororo-SV (Li et al., 2019b), and (2) a single sentence from which models generate multi-scene videos – HiREST (Zala et al., 2023). Coref-SV is a new multi-scene text description dataset that we propose to evaluate the visual consistency of objects across multi-scene videos. We create Coref-SV by augmenting the Pororo-SV dataset (Li et al.,

2019b; Kim et al., 2017), which consists of multi-scene paragraphs from the "Pororo the Little Penguin" animated series. To evaluate the temporal consistency of video generation models trained on real-world videos, we replace its original animation characters (e.g., Pororo) with humans and common animals and examine their appearance across different scenes. Recurring character names are replaced with pronouns (she/he/it). See Appendix D.3 for prompt preparation details.

**Automated evaluation metrics.** Following previous works (Hong et al., 2022; Wu et al., 2022b; Wang et al., 2023b), we use **FID** (Heusel et al., 2017) and **FVD** (Unterthiner et al., 2019) as video quality metrics, and **CLIPSIM** (Wu et al., 2021) as the text-video alignment metric. Given that CLIP fails to faithfully capture detailed semantics such as spatial relations, object counts, and actions in videos (Otani et al., 2023; Cho et al., 2023a;b; Hu et al., 2023), we further propose the use of fine-grained evaluation metrics. For the evaluation of VPEval Skill-based prompts, we employ **VPEval accuracy** based on running skill-specific evaluation programs (object, count, spatial, scale) that execute relevant visual modules (Cho et al., 2023b). For ActionBench-Direction prompts, we propose an **object movement direction accuracy** metric that takes both temporal information and spatial layouts into consideration. To achieve this, we obtain the start/end locations of objects by detecting them with GroundingDINO (Liu et al., 2023) in the first/last video frames. We then evaluate whether the $x$-coordinates (for movements left or right) or $y$-coordinates (for movements up or down) of the objects have changed correctly. For **consistency** evaluation in ActivityNet Captions and Coref-SV, we introduce a new metric to measure the consistency of the visual appearance of a target object across different scenes. For this, we first detect the target object using GroundingDINO from the center frame of each scene video. Then, we extract the CLIP (ViT-B/32) image embedding from the crop of the detected bounding box. We calculate the multi-scene object consistency metric by averaging the CLIP image embedding similarities across all adjacent scene pairs: $\frac{1}{N} \sum_{n=1}^{N-1} cos(\text{CLIP}_n^{\text{img}}, \text{CLIP}_{n+1}^{\text{img}})$, where $N$ is the number of scenes, $cos(\cdot, \cdot)$ is cosine similarity, and $\text{CLIP}_n^{\text{img}}$ is the CLIP image embedding of the target object in $n$-th scene.

**Human evaluation.** We conduct a human evaluation on the multi-scene videos generated by both our VIDEODIRECTORGPT and ModelScopeT2V on the Coref-SV dataset. Since we know the target entity and its co-reference pronouns in the Coref-SV prompts, we can compare the temporal consistency of the target entities across scenes. We evaluate the human preference between videos from two models in each category of Quality, Text-Video Alignment, and Object Consistency. We show 50 videos of each model to three crowd-workers from Amazon Mechanical Turk to rate and then we calculate human preferences between the models. See Appendix E for more setup details.

**Step-by-step error analysis.** We also do an error analysis with an expert on each step of our single-sentence to multi-scene video generation pipeline on the HiREST dataset. We analyze the generated multi-scene text descriptions, layouts, and entity/background consistency groupings to evaluate our video planning stage, and examine the final video to evaluate the video generation stage. We provide the detailed error analysis setup in Appendix E.

## 5 RESULTS AND ANALYSIS

### 5.1 SINGLE-SCENE VIDEO GENERATION

**Layout control results (VPEval Skill-based prompts).** Table 1 (left) displays the VPEval accuracy on the VPEval Skill-based prompts. Our VIDEODIRECTORGPT significantly outperforms ModelScopeT2V on all layout control skills. These results suggest that layouts generated by our LLM are highly accurate and greatly improve the control of object placements during video generation. Fig. 4 (1st row) shows a example where our LLM-generated *video plan* successfully guides Layout2Vid to accurately place the objects. In contrast, ModelScopeT2V fails to generate a 'pizza'.

**Object movement results (ActionBench-Direction).** Table 1 (right) shows the performance on the ActionBench-Direction prompts which evaluate both temporal understanding and spatial layout control. Our VIDEODIRECTORGPT outperforms ModelScopeT2V in object movement direction accuracy by a large margin, demonstrating that our LLM-generated layouts can improve the accuracy of object dynamics in video generation. Fig. 4 (2nd row) shows video generation examples, where our LLM-generated *video plan* guides the Layout2Vid module to place in the correct starting position and guide the 'pear' to the correct end position in the video, whereas the 'pear' in the ModelScopeT2V video moves in a random wrong direction.

Table 1: Comparison of VIDEODIRECTORGPT with ModelScopeT2V on layout control (VPEval Skill-based) and object movement (Actionbench-Direction) for single-scene video generation.

| Method | VPEval Skill-based | | | | | ActionBench-Direction |
|---|---|---|---|---|---|---|
| | Object | Count | Spatial | Scale | Overall Acc. (%) | Movement Direction Acc. (%) |
| ModelScopeT2V | 89.8 | 38.8 | 18.0 | 15.8 | 40.8 | 30.5 |
| VIDEODIRECTORGPT (Ours) | **97.1** | **77.4** | **61.1** | **47.0** | **70.6** | **46.5** |

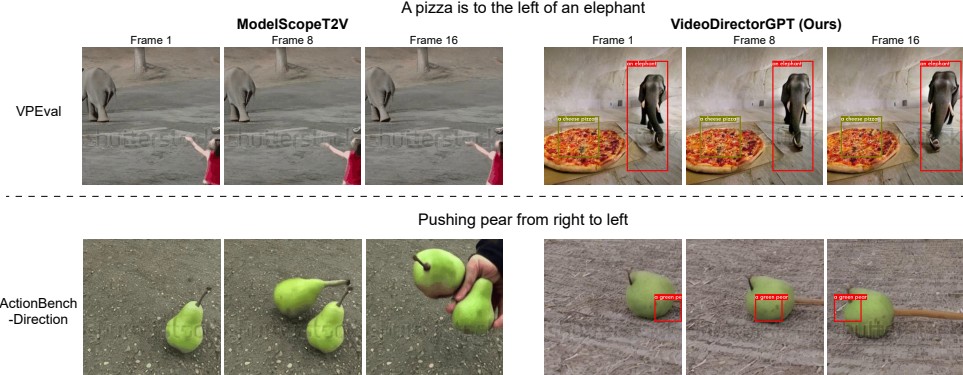

Figure 4: Generation examples on a **VPEval Skill-based prompt** and an **ActionBench-Direction prompt**. Our *video plan*, with object layouts overlaid, successfully guides the Layout2Vid module to place objects in the correct spatial relations for the VPEval Skill-based prompt and move the 'pear' in the correct direction for the ActionBench-Direction prompt, whereas ModelScopeT2V fails to generate a 'pizza' in the VPEval Skill-based prompt example and the 'pear' moves in a random wrong direction for the ActionBench-Direction prompt. See Appendix G for additional examples and supplementary material for full videos.

**Open-domain results (MSR-VTT).** Table 2 shows the visual quality (FVD, FID) and text-video alignment (CLIPSIM) metrics. Our VIDEODIRECTORGPT maintains similar performance as its closest baseline ModelScopeT2V (good improvement in FVD, and similar performance on FID and CLIPSIM), while additionally being equipped with layout control and multi-scene temporal consistency. In addition, our VIDEODIRECTORGPT achieves better or comparable performance to models trained with larger video data (e.g., Make-A-Video) or with higher resolution (e.g., VideoLDM).

Table 2: Visual quality and text-video alignment metrics on MSR-VTT. ModelScopeT2V[†]: Our replication with 2990 randomly selected test prompts.

| Method | Visual quality | | T-V alignment |
|---|---|---|---|
| | FVD ($\downarrow$) | FID ($\downarrow$) | CLIPSIM ($\uparrow$) |
| *Different arch / Training data* | | | |
| NUWA | – | 47.68 | 0.2439 |
| CogVideo (Chinese) | – | 24.78 | 0.2614 |
| CogVideo (English) | 1294 | 23.59 | 0.2631 |
| MagicVideo | 1290 | – | – |
| VideoLDM | – | – | 0.2929 |
| Make-A-Video | – | 13.17 | 0.3049 |
| *Same video backbone & Test prompts* | | | |
| ModelScopeT2V[†] | 683 | 12.32 | **0.2909** |
| VIDEODIRECTORGPT (Ours) | **550** | **12.22** | 0.2860 |

## 5.2 MULTI-SCENE VIDEO GENERATION

**Multiple sentences to multi-scene videos (ActivityNet Captions / Coref-SV).** As shown in the left two blocks of Table 3, our VIDEODIRECTORGPT outperforms ModelScopeT2V in visual quality (FVD/FID) and consistency on ActivityNet Captions and Coref-SV datasets. Notably, for Coref-SV, our VIDEODIRECTORGPT achieves higher object consistency than ModelScopeT2V even with GT co-reference (where pronouns are replaced with their original noun counterparts, acting as oracle information; e.g., "she picked up ..." becomes "cat picked up ..."), showcasing the strong object identity preservation of our framework. Fig. 5 (left) shows a video generation example from Coref-SV, where the LLM-generated *video plan* can guide the Layout2Vid module to generate the same mouse across scenes consistently, whereas ModelScopeT2V generates a hand and a dog instead of a mouse in later scenes. See Appendix G for an additional example.

Table 3: Multi-scene video generation with multiple input sentences (ActivityNet Captions and Coref-SV) and single sentence (HiREST prompts). *GT co-reference*: replacing co-reference pronouns in Coref-SV with the original object names (e.g., "his friends" becomes "dog's friends" if the original object is 'dog').

| Method | ActivityNet Captions | | | Coref-SV | HiREST | |
|---|---|---|---|---|---|---|
| | FVD (↓) | FID (↓) | Consistency (↑) | Consistency (↑) | FVD (↓) | FID (↓) |
| ModelScopeT2V | 980 | 18.12 | 46.0 | 16.3 | 1322 | 23.79 |
| ModelScopeT2V (with GT co-reference; oracle) | - | - | - | 37.9 | - | - |
| VIDEODIRECTORGPT (Ours) | **805** | **16.50** | **64.8** | **42.8** | **733** | **18.54** |

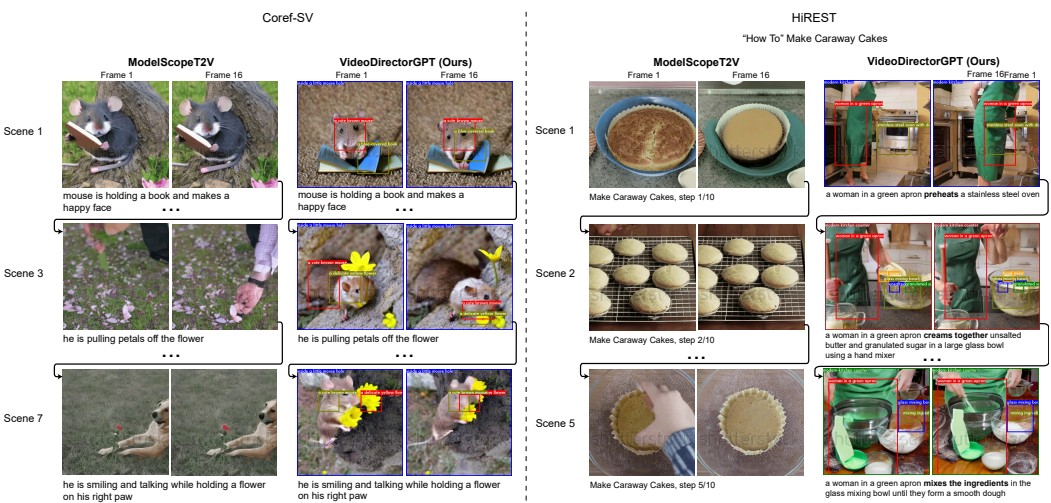

Figure 5: Generation examples on **Coref-SV (left)** and **HiREST (right)**. For both Coref-SV and HiREST, our VIDEODIRECTORGPT is able to generate detailed *video plans* and visually consistent videos. In HiREST, the plan also expands the original text prompt to show the process. Conversely, ModelScopeT2V generates a hand and a dog instead of a mouse for Coref-SV, and only generates the final caraway cake (which is visually inconsistent). More examples are in Appendix G and see supplementary for full videos.

**Single sentence to multi-scene videos (HiREST).** The right block of Table 3 shows our VIDEODI-RECTORGPT achieves better visual quality scores (FVD/FID) than ModelScopeT2V on the HiREST dataset. As shown in Fig. 5 (right), our LLM can generate a step-by-step *video plan* from a single prompt and our Layout2Vid can generate consistent videos following the plan. Our VIDEODIREC-TORGPT generates a step-by-step video showing how to make caraway cakes (a British seed cake). ModelScopeT2V repeatedly generates the final caraway cake (which is visually inconsistent). We include an additional example in Appendix G.

### 5.3 ADDITIONAL ANALYSIS

**Generating videos with custom image exemplars.** Our Layout2Vid can obtain CLIP image embeddings either from user-provided image exemplars or from entity text descriptions via the Karlo Prior. In Fig. 6, we demonstrate that our Layout2Vid can flexibly take either text-only or image+text descriptions as input to generate multi-scene videos with good entity consistency.

**Dynamic layout strength control based on GPT-4.** The number of denoising steps with layout guidance, denoted as $\alpha$ (detailed in Appendix C.3), is a key hyper-parameter in our model. Instead of using a static $\alpha$ value, we explore dynamically adjusting it during the *video plan* generation by asking the LLM how much layout guidance needs to be enforced for each prompt. Table 4 shows the result with static $\alpha$ values of 0.1, 0.2, and 0.3, as well as dynamic $\alpha$ values determined by GPT-4 (called LLM-Dynamic-$\alpha$). Interestingly, LLMs can help the video generation process achieve a good balance in the quality-layout trade-off. Detailed explanation for Table 4 is given in Appendix F.

Table 4: Ablation of the denoising steps with layout guidance (via Guided 2D attentions) in open-domain (MSR-VTT) and object dynamics (ActionBench-Direction) prompts. $\alpha = \frac{\text{\# steps with layout guidance}}{\text{\# total steps}}$. Our Layout2Vid module uses 50 denoising steps in total.

| # Denoising steps with layout guidance | MSR-VTT | | | ActionBench-Direction |
|---|---|---|---|---|
| | FVD ($\downarrow$) | FID ($\downarrow$) | CLIPSIM ($\uparrow$) | Movement Direction Acc. (%) |
| $\alpha = 0.1$ (5 steps) | 550 | **12.22** | **0.2860** | 46.5 |
| $\alpha = 0.2$ (10 steps) | 588 | 17.25 | 0.2700 | **59.8** |
| $\alpha = 0.3$ (15 steps) | 593 | 17.17 | 0.2702 | 57.8 |
| LLM-Dynamic-$\alpha$ (5-15 steps) | **523** | 13.75 | 0.2790 | 56.8 |

Figure 6: Video generation with text-only and image+text inputs. Users can provide either text-only or image+text descriptions to place custom entities when generating videos with VIDEODIRECTORGPT. The identities of the entities are preserved across multiple scenes. Additional examples are shown in Appendix G and see supplementary for full videos.

## 5.4 HUMAN EVALUATION

We conduct a human evaluation (detailed in Sec. 4) on multi-scene videos generated by both VIDEODIRECTORGPT and ModelScopeT2V on the Coref-SV dataset. Table 5 shows that VIDEODIRECTORGPT achieves a higher preference than ModelScopeT2V in all categories (Quality, Text-Video Alignment, and Object Consistency).

Table 5: Human preference on generated multi-scene videos of Coref-SV in three evaluation categories.

| Evaluation category | Human Preference (%) $\uparrow$ | | |
|---|---|---|---|
| | VIDEODIRECTORGPT (Ours) | ModelScopeT2V | Tie |
| Quality | **54** | 34 | 12 |
| Text-Video Alignment | **54** | 28 | 18 |
| Object Consistency | **58** | 30 | 12 |

We also conduct an error analysis on our single-sentence to multi-scene video generation pipeline on HiREST prompts and find that our LLM-guided planning steps score high accuracy, whereas the biggest score drop happens in the layout-guided video generation. This suggests that our VIDEODIRECTORGPT could generate more accurate videos, once we have access to a stronger T2V backbone than ModelScopeT2V. We present full error analysis results in Appendix E.

## 6 CONCLUSION

In this work, we propose VIDEODIRECTORGPT, a novel framework for consistent multi-scene video generation, leveraging the knowledge of LLMs for video content planning and grounded video generation. In the first stage, we employ GPT-4 as a video planner to craft a *video plan*, which provides a multi-component script for videos with multiple scenes. In the second stage, we introduce Layout2Vid, a grounded video generation module, to generate videos with layout and cross-scene consistency control. Our experiments demonstrate that our proposed VIDEODIRECTORGPT framework substantially improves object layout and movement control and can generate multi-scene videos with cross-scene visual consistency, while achieving competitive performance with SOTAs on open-domain single-scene T2V generation.

## 7 ETHICS STATEMENT

While our framework can be beneficial for numerous applications (e.g., user-controlled/human-in-the-loop video generation/manipulation and data augmentation), akin to other video generation frameworks, it can also be utilized for potentially harmful purposes (e.g., creating false information or misleading videos), and thus should be used with caution in the real-world applications (with human supervision). Our video generation module (Layout2Vid) is based on the pretrained weights of ModelScopeT2V. Therefore, we face similar limitations to their model, including deviations related to the distribution of training datasets, imperfect generation quality, and only understanding the English corpus.

## 8 REPRODUCIBILITY STATEMENT

Our model is built upon the publicly available code repository from GLIGEN (Li et al., 2023)[1] and ModelScopeT2V (Wang et al., 2023b)[2]. Please see Sec. 3/Appendix C for model architecture details, Sec. 4/Appendix D.2/Appendix D.3 for dataset details, and Sec. 4/Appendix D.4 for metric details. We will publicly release our code.

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

# Appendix

## A ADDITIONAL RELATED WORKS

**Text-to-video generation.** The text-to-video (T2V) generation task is to generate videos from text descriptions. Early T2V generation models (Li et al., 2017; 2019b) used variational autoencoders (VAE) (Kingma & Welling, 2014) and generative adversarial networks (GAN) (Goodfellow et al., 2020), while multimodal language models (Hong et al., 2022; Wu et al., 2022a; Villegas et al., 2023; Maharana et al., 2022; Ge et al., 2022; Wu et al., 2021) and denoising diffusion models (Ho et al., 2022; Singer et al., 2023; Blattmann et al., 2023; Khachatryan et al., 2023; Wang et al., 2023a; Yin et al., 2023) have become popular for recent works. Since training a T2V generation model from scratch is computationally expensive, recent work often leverages pre-trained text-to-image (T2I) generation models such as Stable Diffusion (Rombach et al., 2022) by finetuning them on text-video pairs (Wang et al., 2023b; Blattmann et al., 2023). While this warm-start strategy enables high-resolution video generation, it comes with the limitation of only being able to generate short video clips, as T2I models lack the ability to maintain consistency through long videos. Recent works on long video generation (Blattmann et al., 2023; Yin et al., 2023; Villegas et al., 2023; He et al., 2023) aim at generating long videos of a few minutes. However, the generated videos often display the continuation or repetitive patterns of a single action (e.g., driving a car) instead of transitions and dynamics of multiple changing actions/events (e.g., five steps about how to bake a cake). In this work, we address this problem of multi-scene video generation with a two-stage framework: using an LLM (e.g., GPT-4) to generate a structured *video plan* (consisting of stepwise scene descriptions, entities and their layouts) and generating videos using Layout2Vid, a layout-guided text-to-video generation model with consistency control. Our Layout2Vid infuses layout control and multi-scene temporal consistency into a pretrained T2V generation model via data and parameter-efficient training, while preserving its original visual quality.

## B VIDEO PLANNING

**GPT-4 prompt templates.** In this section, we provide the prompt templates we give to our video planner (Sec. 3.1). The *video plan* is generated in two steps by prompting GPT-4[3] with different in-context examples (we use 1 and 5 in-context examples for the first and second steps, respectively). In the first step (see Fig. 7), we ask GPT-4 to expand a single text prompt into a multi-scene *video plan*. Each scene comes with a text description, a list of entities, and a background. In the second step (see Fig. 8), we generate a list of bounding boxes for the entities in each frame based on the list of entities and the scene description. In line with VPGen (Cho et al., 2023b), we utilize the $[x_0, y_0, x_1, y_1]$ format for bounding boxes, where each coordinate is normalized to fall within the range [0,1]. For in-context examples, we present 0.05 as the minimum unit for the bounding box, equivalent to a 20-bin quantization over the [0,1] range.

**API cost.** Using GPT-4 tokenizer, the average input/output token lengths of each step are 2K/1K for the first step and 6K/1K for the second step. Using GPT-4, it takes 0.12 USD and 0.24 USD for the inference of the first and second steps, respectively.

---

**Instructions:**
Given a single text prompt, you need to envision a multi-scene video by generating a sequence of stepwise prompts to describe the text prompt. For each step, you also need to generate the set of entities needed and describe the background scene where the video should occur. Related steps should maintain similar entities and background scenes. Before you write each stepwise description, you must follow these instructions:
1. Each step prompt must contain only a single motion or action.
2. Each step prompt must include all relevant objects and describe the environment scene.
3. Make sure each step prompt must be easy described by a shot video cilp of 8 seconds at 2fps.
Use the following format: [[step_1: [prompt, entities, background]; step_2: [prompt, entities, background]; … step_n: [prompt, entities, background]]

**In-context example:**
**Input:** Provide stepwise prompts for the text prompt: make butter biscuits
**Output:**
Step 1: [Prompt: A chief preheats the oven; Entities: chief, oven; Background: kitchen counter];
Step 2: [Prompt: A chief creams together butter and sugar in a mixing bowl; Entities: chief, mixing bowl, butter, sugar, hand mixer; Background: kitchen]
Step 3: [Prompt: A chief beats in eggs into the mixing bowl; Entities: chief, eggs, mixing bowl; Background: kitchen];
  …

**Input text:** Provide stepwise prompts for the text prompt: [*insert single text prompt here*]

Figure 7: Prompt template for the 1st video planning step (scene descriptions and entities/background generation).

## C LAYOUT2VID

### C.1 PRELIMINARIES: MODELSCOPET2V FOR T2V GENERATION.

We implement Layout2Vid by injecting layout control capability into ModelScopeT2V (Wang et al., 2023b), a public text-to-video generation model based on Stable Diffusion (Rombach et al., 2022). ModelScopeT2V consists of (1) a CLIP ViT-H/14 (Radford et al., 2021) text encoder, (2) an autoencoder, and (3) a diffusion UNet (Ronneberger et al., 2015; Ho et al., 2020). Given a $T$ frame video $\boldsymbol{x} \in \mathbb{R}^{T \times 3 \times H \times W}$ with video caption $\boldsymbol{c}$ and frame-wise layouts $\{\boldsymbol{e}\}_{i=1}^{T}$, ModelScopeT2V first uses an autoencoder to encode the video into a latent representation $\boldsymbol{z} = \mathcal{E}(\boldsymbol{x})$. The diffusion UNet performs denoising steps in the latent space to generate videos, conditioned on the CLIP text encoder representation of video captions. The UNet comprises a series of spatio-temporal blocks, each containing four modules: spatial convolution, temporal convolution, spatial attention, and temporal attention. Since the original ModelScopeT2V does not offer control beyond the text input, we build upon the 2D attention module in the spatial attention module to create 'Guided 2D Attention', which allows for spatial control using bounding boxes.

### C.2 TRAINING AND INFERENCE DETAILS

The highlight of our Layout2Vid training is that it was conducted solely on image-level data with bounding box annotations. We trained the MLP layers for grounding tokens and the Guided 2D Attention layer with the same bounding-box annotated data used in GLIGEN (Li et al., 2023), which

---

[3]We employ the `gpt-4-0613` version via OpenAI API.

Instructions:
Assuming the frame size is normalized to the range 0-1, you need to give a possible 8-frame layout with bounding boxes of the listed entities of a given scene description. Each object in the image is one rectangle or square box in the layout and size of boxes should be as large as possible. You need to generate layouts from the close up camera view of the event. The layout difference between two adjacent frames must be small, considering the small interval. You also need to generate a caption that best describes the image for each frame. After generating all frames, add reasoning to your design. Use format:
Frame_1: [[object1, [left, top, right, bottom]], [object2, [left, top, right, bottom]], ..., [object_n, [left, top, right, bottom]]], caption:...
Frame_2: [[object1, [left, top, right, bottom]], [object2, [left, top, right, bottom]], ..., [object_n, [left, top, right, bottom]]], caption:...
...
Frame_8: [[object1, [left, top, right, bottom]], [object2, [left, top, right, bottom]], ..., [object_n, [left, top, right, bottom]]], caption:...
Reasoning:...

In-context example:
**Input:** Please generate bounding box coordinates for the following entities based on the scene description.
Entities: ['table', 'cat', 'chair']. Scene description: A cat jumps from a chair to a table.
**Output:**
Frame_1: [["table", [0.1, 0.6, 0.9, 1.0]], ["cat", [0.4, 0.55, 0.6, 0.6]], ["chair", [0.05, 0.6, 0.4, 1.0]]], caption: A cat is standing on the table, preparing to jump.
Frame_2: [["table", [0.1, 0.6, 0.9, 1.0]], ["cat", [0.45, 0.5, 0.65, 0.55]], ["chair", [0.05, 0.6, 0.4, 1.0]]], caption: The cat has just left the table and is in mid-air.
...
Reasoning: The cat\'s bounding box in Frame 1 is on the table, showing the cat in the take-off position. From Frame 2 to Frame 6, the cat is shown in various stages mid-flight, with the cat\'s bounding box moving closer to the chair with each frame. ...

**Input text:** Provide bounding box coordinates for the prompt: [*insert entities and scene description here*]

Figure 8: Prompt template for the 2nd video planning step (layout generation).

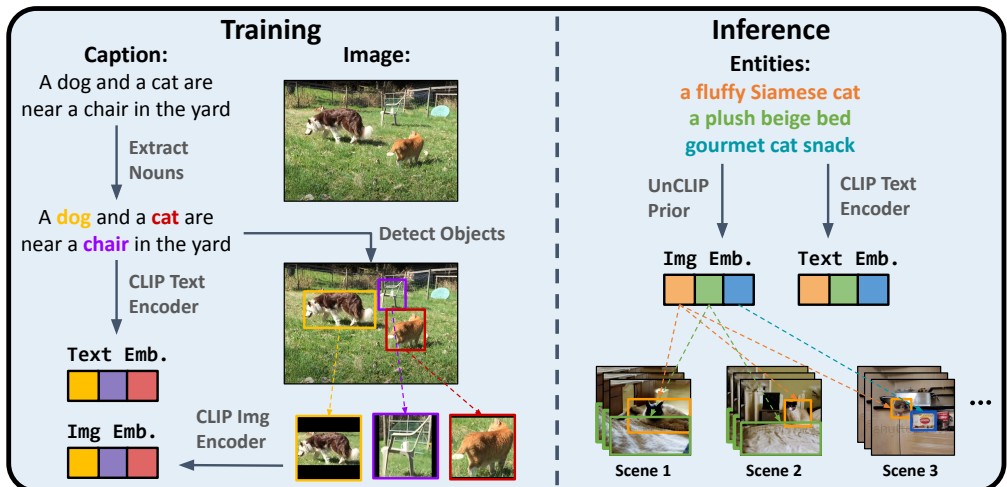

Figure 9: **Training and Inference procedure of Layout2Vid.** A notable aspect of our model's training is that it was conducted only on image-level data with bounding box annotations. During training, we extract entities (as noun phrases) from the video caption and apply object detectors to pinpoint the bounding box locations of these entities. Then we obtain image embeddings for the entities by encoding the image crop of the entities with CLIP image encoder. During inference, we apply unCLIP Prior on the entities generated by the LLM to retrieve their corresponding CLIP image embeddings. To preserve visual consistency, the same joint image-text embedding pair is used across scenes to represent an identical object.

consists of 0.64M images. We train Layout2Vid for 50k steps, which takes only two days with 8 A6000 GPUs (each 48GB memory). All the remaining modules in the spatio-temporal block Fig. 3 are frozen during the training phase. We illustrate the training and inference procedure of Layout2Vid in Fig. 9.

During training, we first use spaCy (Honnibal & Montani, 2017) to extract all noun phrases from the video caption, and use CLIP text encoder to get their text embeddings. Then we apply GroundingDINO (Liu et al., 2023) to detect corresponding bounding box locations. To remove redundant and duplicate bounding boxes, we only keep the bounding boxes that overlap less than 95% with other boxes. Next, we crop the bounding box areas and resize the cropped images so that the longest edge has size 224. We pad the cropped images with black colors to transform them into square shapes. Finally, the CLIP ViT-L/14 image encoder converts cropped images to embeddings. We use

the joint image-text embeddings for grounding token construction. Our training can be completed in only 2 days on a server with 8 A6000 GPUs.

During inference, we use the Karlo implementation of unCLIP Prior to the entities to convert the texts into their corresponding CLIP image embeddings, and CLIP text encoder to get their corresponding text embeddings. We use CLIP ViT-L/14 as a backbone during training to be consistent with Karlo. This helps us to preserve the visual consistency of the same object by using the same image embedding across scenes.

### C.3 LAYOUT-GUIDED DENOISING STEPS.

During video generation, we use two-stage denoising in Layout2Vid following (Li et al., 2023), where we first use layout-guidance with Guided 2D attention for $\alpha * N$ steps, and use the denoising steps without Guided 2D attention for the remaining $(1 - \alpha) * N$ steps, where $N$ is the total number of denoising steps, and $\alpha \in [0, 1]$ is the ratio of layout-guidance denoising steps. In our ablation study (Appendix F), we find that a high $\alpha$ could increase layout control but lead to lower visual quality, which is also consistent with the finding in Li et al. (2023). By default, we use $\alpha = 0.1$ and $N = 50$ denoising steps. We also explore using the LLM to determine the $\alpha$ value within the range $[0, 0.3]$ during the *video plan* generation (see Appendix F for details).

## D    EXPERIMENT SETUP

We provide additional details on our experiment setups (from Sec. 4) below.

### D.1    EVALUATED MODELS

We compare our VIDEODIRECTORGPT to 6 popular T2V generation models, NUWA (Wu et al., 2022b), CogVideo (Hong et al., 2022), VideoLDM (Blattmann et al., 2023), MagicVideo (Zhou et al., 2022), Make-A-Video (Singer et al., 2023), and ModelScopeT2V (Wang et al., 2023b). Since NUWA, VideoLDM, MagicVideo, Make-A-Video, and CogVideo (English) are not publicly available, we primarily compare our VIDEODIRECTORGPT with ModelScopeT2V, and present comparisons with the other models on the datasets for which their papers have provided results. ModelScopeT2V is the closest baseline to our framework among all these models, because our Layout2Vid utilizes its frozen weights and only trains a small set of new parameters to add spatial control and temporal consistency across multiple scenes.

### D.2    PROMPTS FOR SINGLE-SCENE VIDEO GENERATION

For single-scene video generation, we conduct experiments with VPEval Skill-based prompts to evaluate layout control (Cho et al., 2023b), ActionBench-Direction prompts to assess object dynamics (Wang et al., 2023c), and MSR-VTT to cover diverse open-domain scenes (Xu et al., 2016).

**VPEval Skill-based prompts** evaluate different object-centric layout control skills in text-to-image/video generation. We randomly sample 100 prompts for each of the four skills: Object (generation of a single object), Count (generation of a specific number of objects), Spatial (generation of two objects with a spatial relation; e.g., left/right/above/below), and Scale (generation of two objects with a relative scale relation; e.g., bigger/smaller/same).

**ActionBench-Direction prompts** evaluate the action dynamics (object movement directions) in video language models. We prepare the prompts by sampling video captions from ActionBench-SSV2 (Wang et al., 2023c) and balancing the distribution of movement directions. Concretely, we select captions from the ActionBench-SSV2 validation split that include phrases like 'right to left' or 'left to right' (e.g., 'pushing a glass from left to right'), which are common phrases describing movement directions in the captions. Then we augment these prompts by switching the directions to each of four directions: 'left to right', 'right to left', 'top to bottom', and 'bottom to top' to create 100 prompts for each direction. We call the resulting 400 prompts as *ActionBench-Direction* prompts. These prompts ensure a balanced distribution of movement directions while maintaining diversity in objects.

**MSR-VTT** is an open-domain video captioning dataset, which allows us to check if our Layout2Vid maintains the original visual quality and text-video alignment performance of the ModelScopeT2V backbone after integration of the layout/movement control capabilities. The MSR-VTT test set comprises 2,990 videos, each paired with 20 captions. Following VideoLDM (Blattmann et al., 2023), we sample one caption from the 20 available captions for each video and use the 2,990 corresponding generated videos for evaluation.

### D.3 PROMPTS FOR MULTI-SCENE VIDEO GENERATION

For multi-scene video generation, we experiment with two types of input prompts: (1) a list of sentences describing events – ActivityNet Captions (Krishna et al., 2017) and Coref-SV prompts based on Pororo-SV (Li et al., 2019b) and (2) a single sentence from which models generate multi-scene videos – HiREST (Zala et al., 2023).

**ActivityNet Captions** is a dense-captioning dataset designed for detecting and describing multiple events in videos using natural language. For the multi-scene video generation task, we use 165 randomly sampled videos from the validation split and use the event captions as input for ModelScopeT2V and our VIDEODIRECTORGPT. When calculating object consistency (see Appendix D.4), we find the subject of the first event caption (via spaCy dependency parser (Honnibal & Montani, 2017)) and check its appearance in multiple scenes.

**Coref-SV** is a new multi-scene text description dataset that we propose to evaluate the consistency of object appearances across multi-scene videos. We prepare the Coref-SV prompts by augmenting the Pororo-SV dataset (Li et al., 2019b; Kim et al., 2017), which consists of multi-scene paragraphs from the "Pororo the Little Penguin" animated series. To evaluate the temporal consistency of video generation models trained on real-world videos, we extend its original animation characters (e.g., Pororo) to humans and common animals and examine their appearance across different scenes. Concretely, we sample 10 episodes, each consisting of multiple scenes (6.2 scenes per episode on average). Then, we replace the first appearance of a character with one of the predefined 10 real-world entities (e.g., person/dog, etc.) and replace the remaining appearances of the character with pronouns (e.g., he/she/it/etc.). In total, we obtain 100 episodes (=10 episodes $\times$ 10 entities) in Coref-SV. In order to generate visually consistent entities, the multi-scene video generation models would need to address the co-reference of these target entities across different scenes. We use the final scene descriptions as input for both ModelScopeT2V and VIDEODIRECTORGPT. When calculating object consistency (see Appendix D.4), we use the selected entity as the target object.

**HiREST** provides step annotations for instructional videos paired with diverse 'How to' prompts (e.g., a video paired with 'how to make butter biscuits' prompt is broken down into a sequence of short video clips of consecutive step-by-step instructions). For the multi-scene video generation task, we employ 175 prompts from the test splits, where we only include the prompts with step annotations, to ensure that it is possible to create multi-scene videos from the prompts. Note that instead of providing a list of scene description sentences like in ActivityNet Captions/Coref-SV, we only give the single high-level 'How to' prompt and let the models generate a multi-scene video from it. In VIDEODIRECTORGPT, our LLM can automatically generate the multi-scene *video plan* and video from the input prompt. In contrast, for the ModelScopeT2V baseline, we help the model understand the different number of scenes to generate by pre-defining the number of scenes $N$, and independently generate $N$ videos by appending the suffix "step $n/N$" to the prompt for $n$-th scene (e.g., "Cook Beet Greens, step 1/10"). To ensure that our VIDEODIRECTORGPT videos and ModelScopeT2V videos are equal in length, we use the same number of scenes generated by our LLM during the planning stage for ModelScopeT2V.

### D.4 AUTOMATED EVALUATION METRICS

**Visual quality and text-video alignment.** Following previous works (Hong et al., 2022; Wu et al., 2022b; Wang et al., 2023b), we use Fréchet Inception Distance (FID) (Heusel et al., 2017), with InceptionV3 (Szegedy et al., 2016) as the backbone, and Fréchet Video Distance (FVD) (Unterthiner et al., 2019), with I3D (Carreira & Zisserman, 2017) as the backbone, for video quality metrics. Additionally, we use CLIPSIM (Wu et al., 2021) (with CLIP ViT-B/16 (Radford et al., 2021)) for the text-video alignment metric. Given that CLIP fails to faithfully capture detailed semantics such

as spatial relations, object counts, and actions in videos (Otani et al., 2023; Cho et al., 2023a;b; Hu et al., 2023), we further propose the use of the following fine-grained evaluation metrics.

**VPEval accuracy.** For the evaluation of VPEval Skill-based prompts, we employ VPEval accuracy based on running skill-specific (object, count, spatial, scale) evaluation programs that execute relevant visual modules (Cho et al., 2023b). For each video, we uniformly sample 4 frames and average the frame-level VPEval accuracy to obtain the final score.

**Object movement direction accuracy.** Since the VPEval accuracy described above does not cover temporal information, we propose a metric that takes into account temporal information as well as spatial layouts for ActionBench-Direction prompts. To accomplish this, we assess whether the target objects in the generated videos move in the direction described in the prompts. We obtain the start/end locations of objects by detecting objects with GroundingDINO (Liu et al., 2023) on the first/last video frames. We then evaluate whether the $x$ (for movements left or right) or $y$ (for movements up or down) coordinates of the objects have changed correctly and assign a binary score of 0 or 1 based on this evaluation. For instance, given the prompt "pushing a glass from left to right" and a generated video, we identify a 'glass' in both the first and last video frames. We assign a score of 1 if the $x$-coordinate of the object increases by the last frame; otherwise, we assign a score of 0.

**Multi-scene object consistency.** We also introduce a new metric to measure the consistency of the visual appearance of a target object across different scenes. For this, we first detect the target object using GroundingDINO from the center frame of each scene video. Then, we extract the CLIP (ViT-B/32) image embedding from the crop of the detected bounding box. We calculate the multi-scene object consistency metric by averaging the CLIP image embedding similarities across all adjacent scene pairs: $\frac{1}{N} \sum_{n=1}^{N-1} cos(\text{CLIP}_n^{\text{img}}, \text{CLIP}_{n+1}^{\text{img}})$, where $N$ is the number of scenes, $cos(\cdot, \cdot)$ is cosine similarity, and $\text{CLIP}_n^{\text{img}}$ is the CLIP image embedding of the target object in $n$-th scene.

## E  HUMAN EVALUATION DETAILS

We provide details of our human evaluation and error analysis described in Sec. 4. We also show the error analysis results described in Sec. 5.4.

**Human evaluation details.** We conduct a human evaluation study on the multi-scene videos generated by both our VIDEODIRECTORGPT and ModelScopeT2V on the Coref-SV dataset. Since we know the target entity and its co-reference pronouns in the Coref-SV prompts, we can compare the temporal consistency of the target entities across scenes. We evaluate the human preference between videos from two models in each category of Quality, Text-Video Alignment, and Object Consistency. *Quality* measures how well the video looks visually. *Text-Video Alignment* assesses how accurately the video adheres to the input sentences. *Object Consistency* evaluates how well the target object maintains its visual consistency across scenes. We show 50 videos to three crowd-workers from AMT[4] to rate each video and calculate human preferences for each video with average ratings. To ensure high-quality annotations, we require they have an AMT Masters, have completed over 1000 HITs, have a greater than 95% approval rating, and are from one of the United States, Great Britain, Australia, or Canada (as our task is written in the English language). We pay workers $0.06 to evaluate a video (roughly $12-14/hr).

**Step-by-step error analysis details.** We do an error analysis on each step of our single sentence to multi-scene video generation pipeline for HiREST prompts. We analyze the generated multi-scene text descriptions, layouts, and entity/background consistency groupings to evaluate our video planning stage and the final video to evaluate the video generation stage. *Multi-Scene Text Descriptions Accuracy*: we measure how well these descriptions depict the intended scene from the original prompt (e.g., if the original prompt is "Make buttermilk biscuits" the descriptions should describe the biscuit-making process and not the process for pancakes). *Layout Accuracy*: we measure how well the generated layouts showcase a scene for the given multi-scene text description (e.g., the bounding boxes of ingredients should go into a bowl, pan, etc. instead of randomly moving across

---

[4]Amazon Mechanical Turk: `https://www.mturk.com`

the scene). *Entity/Background Consistency Groupings Accuracy*: we measure how well the generated entities and backgrounds are grouped (e.g., entities/backgrounds that should look consistent throughout the scenes should be grouped together). *Final Video Accuracy*: we measure how well the generated video for each scene matches the multi-scene text description (e.g., if the multi-scene text description is "preheating an oven", the video should accurately reflect this). We ask an expert annotator to rank the generations (multi-scene text description, layouts, etc.) on a Likert scale of 1-5 for 50 prompts/videos. Analyzing the errors at each step enables us to check which parts are reliable and which parts need improvement. As a single prompt/video can contain many scenes, to simplify the process for layout and final video evaluation of a prompt/video, we sub-sample three scene layouts and corresponding scene videos and average their scores to obtain the "Layout Accuracy" and "Final Video Accuracy."

**Step-by-step error analysis results.** As shown in Table 6, our LLM-guided planning scores high accuracy on all three components (up to 4.51), whereas the biggest score drop happens in the layout-guided video generation (4.51 → 3.61). This suggests that our VIDEODIRECTORGPT could generate even more accurate videos, once we have access to a stronger T2V backbone than ModelScopeT2V.

Table 6: Step-wise error analysis of VIDEODIRECTORGPT video generation pipeline on HiREST prompts. We use a Likert scale (1-5) to rate the accuracy of the generated components at each step.

| Stage 1: Video Planning (with GPT-4) | | | Stage 2: Video Generation (with Layout2Vid) |
|---|---|---|---|
| Multi-scene Text Descriptions (↑) | Layouts (↑) | Entity/Background Consistency Groupings (↑) | Final Video (↑) |
| 4.92 | 4.69 | 4.52 | 3.61 |

# F  ABLATION STUDIES

In this section, we provide ablation studies on our design choices, including the number of layout-guided denoising steps, different embeddings for layout groundings, and layout representation formats.

**Number of denoising steps with layout guidance.** In Table 4, we show the ablation experiment results of the number of denoising steps with layout guidance during video generation (Sec. 3.2) on MSR-VTT and ActionBench-Direction prompts. For MSR-VTT, we use the same set of randomly sampled test prompts as presented in Table 2. We find that increasing the $\alpha$ ($= \frac{\text{\# steps with layout guidance}}{\text{\# total steps}}$) from 0.1 to 0.2 or 0.3 drops the visual quality (FVD/FID) and text-video alignment (CLIPSIM) on MSR-VTT, while improves the movement direction accuracy in ActionBench-Direction. This quality-layout trade-off is consistent with the finding in layout-guided text-to-image generation models like GLIGEN (Li et al., 2023), where they also found that high $\alpha$ leads to lower visual quality. We also explore dynamically finding the $\alpha$ value for each example during the *video plan* generation by asking the LLM how much layout guidance needs to be enforced for each prompt, instead of using a static $\alpha$ value. As shown in the bottom row ('LLM-Dynamic-$\alpha$') of Table 4, interestingly, LLMs can help the video generation process to have a good balance of quality-layout trade-off.

**Entity grounding embeddings.** As discussed in Sec. 3.2, we compare using different embeddings for entity grounding on 1000 randomly sampled MSR-VTT test prompts. As shown in Table 7, CLIP image embedding is more effective than CLIP text embedding, and using the CLIP image-text joint embedding yields the best results. Thus, we propose to use the image+text embeddings for the default configuration.

**Layout control: bounding box v.s. center point.** In Table 8, we compare different layout representation formats on 1000 randomly sampled MSR-VTT test prompts. We use image embedding for entity grounding and $\alpha = 0.2$ for layout control. Compared with no layout ('w/o Layout input') or center point-based layouts (without object shape, size, or aspect ratio), the bounding box based layout guidance gives better visual quality (FVD/FID) and text-video alignment (CLIPSIM).

Table 7: Ablation of entity grounding embeddings of our Layout2Vid on MSR-VTT and Coref-SV.

| Entity Grounding | MSR-VTT | | | Coref-SV |
|---|---|---|---|---|
| | FVD ($\downarrow$) | FID ($\downarrow$) | CLIPSIM ($\uparrow$) | Consistency (%) |
| Image Emb. | 737 | 18.38 | 0.2834 | 42.6 |
| Text Emb. | 875 | 23.18 | 0.2534 | 36.9 |
| Image+Text Emb. (default) | **606** | **14.60** | **0.2842** | **42.8** |

Table 8: Ablation of layout representation of our VIDEODIRECTORGPT on MSR-VTT. We use $\alpha = 0.2$ and CLIP image embedding for entity grounding.

| Layout representation | FVD ($\downarrow$) | FID ($\downarrow$) | CLIPSIM ($\uparrow$) |
|---|---|---|---|
| w/o Layout input | 639 | 15.28 | 0.2777 |
| Center point | 816 | 18.65 | 0.2707 |
| Bounding box (default) | **606** | **14.60** | **0.2842** |

## G   ADDITIONAL QUALITATIVE EXAMPLES

**VPEval Skill-based.**   Fig. 10 displays generated videos where our LLM-generated video plan successfully guides the Layout2Vid module to accurately place objects in the correct spatial relations and to generate the correct number of objects. In contrast, ModelScopeT2V fails to generate a 'pizza' in the first example and overproduces the number of frisbees in the second example.

**ActionBench-direction.**   Fig. 11 shows video generation examples, where our LLM-generated video plan can guide the Layout2Vid module to place the 'stuffed animal' and the 'pear' in their correct starting positions and then move them toward the correct end positions, whereas the objects in the ModelScopeT2V videos stay in the same location or move in random directions.

**Coref-SV.**   We show another example of our VIDEODIRECTORGPT compared to ModelScopeT2V on a Coref-SV prompt in Fig. 12. Our *video plan* can guide the Layout2Vid module to generate the same dog and maintain snow across scenes consistently, whereas ModelScopeT2V generates different dogs in different scenes and loses the snow after the first scene.

**HiREST.**   We show another example of our VIDEODIRECTORGPT compared to ModelScopeT2V on a HiREST prompt in Fig. 13. Our LLM can generate step-by-step *video plan* from a single prompt and our Layout2Vid can generate consistent videos following the plan. Our VIDEODIRECTORGPT breaks down the process and generates a complete video showing how to make peach melba (a type of dessert consisting of vanilla ice cream and peaches). ModelScopeT2V repeatedly generates the final dessert (which is also inconsistent between scenes).

**Generating videos with custom image exemplars.**   In Fig. 14, we demonstrate that our Layout2Vid can flexibly take either text-only or image+text descriptions as input to generate multi-scene videos with good entity consistency.

## H   LIMITATIONS

While our framework can be beneficial for numerous applications (e.g., user-controlled/human-in-the-loop video generation/manipulation and data augmentation), akin to other video generation frameworks, it can also be utilized for potentially harmful purposes (e.g., creating false information or misleading videos), and thus should be used with caution in real-world applications (with human supervision). Also, generating a *video plan* using the strongest LLM APIs can be costly, similar to other recent LLM-based frameworks. We hope that advances in quantization/distillation and open-source models will continue to lower the inference cost of LLMs. Lastly, our video generation module (Layout2Vid) is based on the pretrained weights of ModelScopeT2V. Therefore, we face similar limitations to their model, including deviations related to the distribution of training datasets, imperfect generation quality, and only understanding the English corpus.

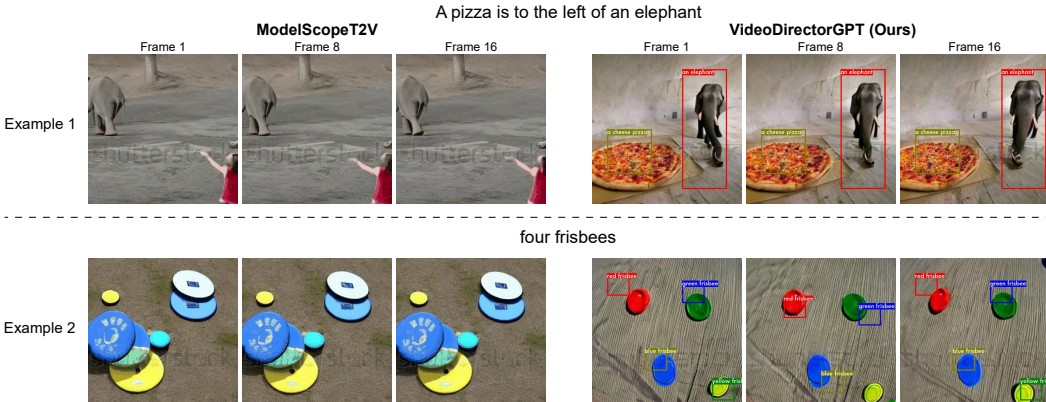

Figure 10: Video generation examples on **VPEval Skill-based prompts** for spatial and count skills. Our *video plan*, with object layouts overlaid, successfully guides the Layout2Vid module to place objects in the correct spatial relations and to depict the correct number of objects, whereas ModelScopeT2V fails to generate a 'pizza' in the first example and overproduces the number of frisbees in the second example. See supplementary material for full videos.

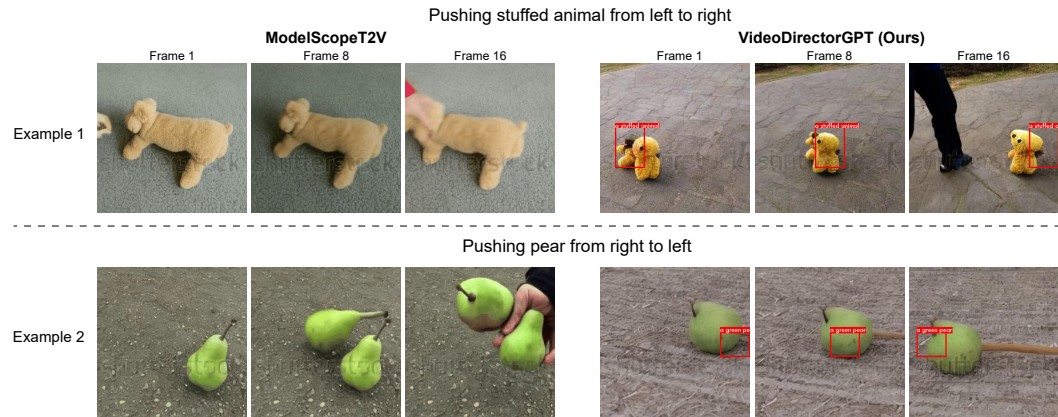

Figure 11: Video generation examples on **ActionBench-Direction prompts**. Our *video plan*'s object layouts (overlaid) can guide the Layout2Vid module to place and move the 'stuffed animal' and 'pear' in their correct respective directions, whereas the objects in the ModelScopeT2V videos stay in the same location or move in random directions. See supplementary material for full videos.

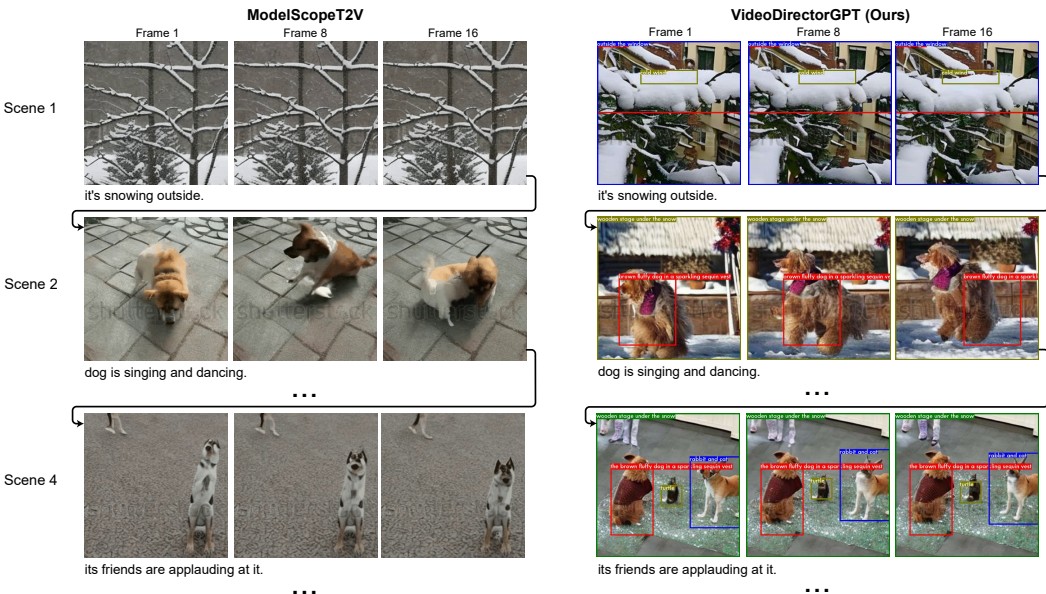

Figure 12: Video generation examples on a **Coref-SV prompt**. Our *video plan*'s object layouts (overlaid) can guide the Layout2Vid module to generate the same brown dog and maintain snow across scenes consistently, whereas ModelScopeT2V generates different dogs in different scenes and loses the snow after the first scene. See supplementary material for the full video.

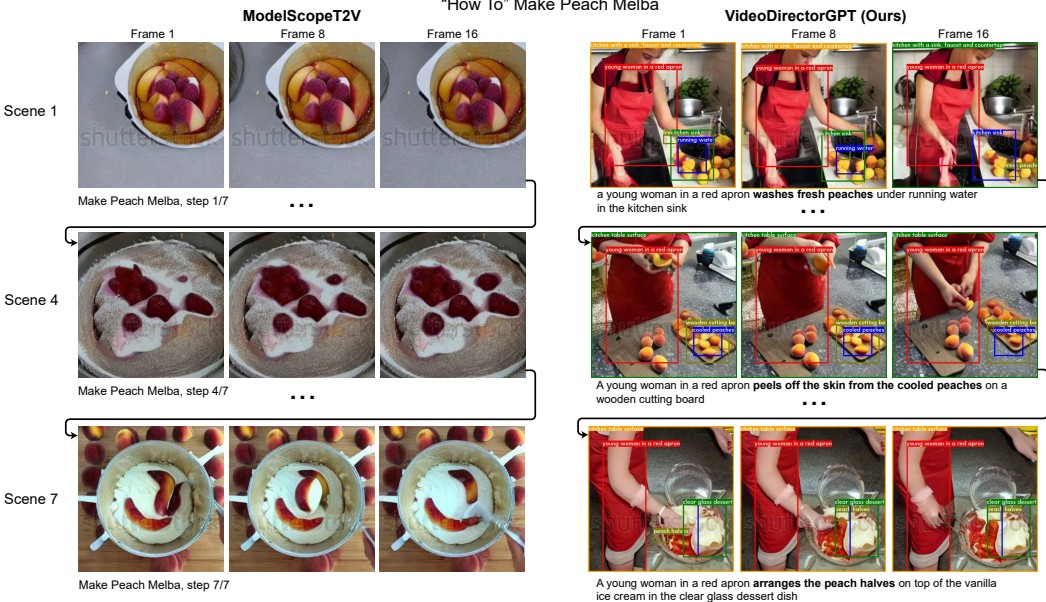

Figure 13: Comparison of generated videos on a **HiREST prompt**. Our VIDEODIRECTORGPT generates a detailed *video plan* that properly expands the original text prompt, ensures accurate object bounding box locations (overlaid), and maintains consistency of the person across the scenes. ModelScopeT2V only generates the final dessert which is not consistent across scenes. See supplementary material for the full video.

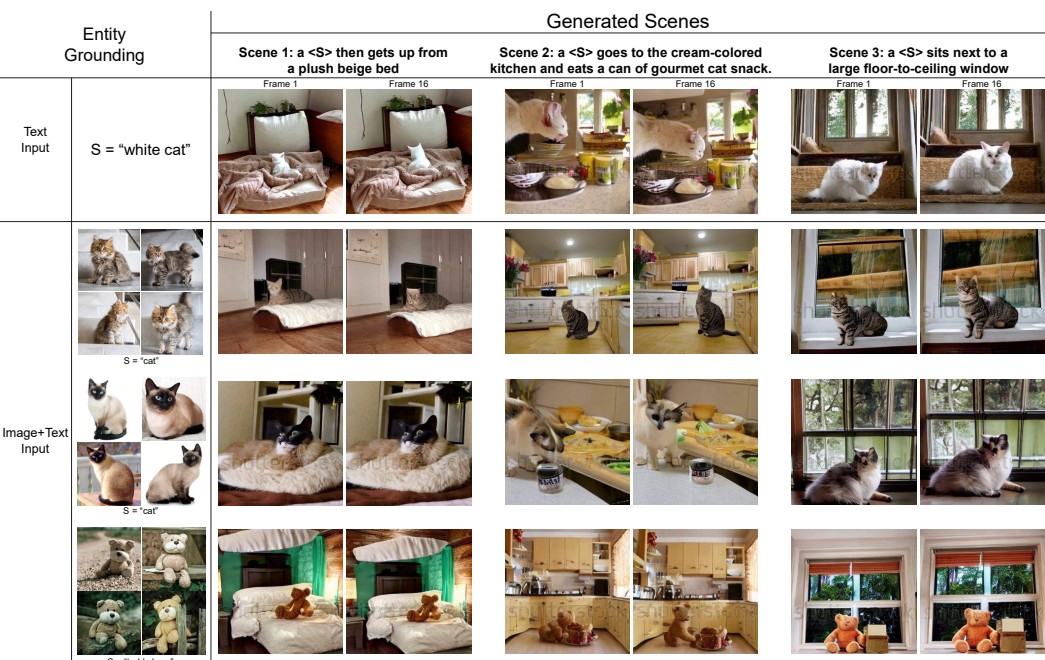

Figure 14: Video generation examples with **text-only and image+text inputs**. Users can flexibly provide either text-only or image+text descriptions to place custom entities when generating videos with VIDEODIRECTORGPT. For both text and image+text based entity grounding examples, the identities of the provided entities are well preserved across multiple scenes. See supplementary material for full videos.

