# OpenReview forum: "VideoDirectorGPT: Consistent Multi-Scene Video Generation via LLM-Guided Planning"
_ICLR.cc/2024/Conference — ICLR 2024 Conference Withdrawn Submission_

### Official Review · Reviewer_dMSD · 2023-11-01

**Soundness:** 3 good
**Presentation:** 3 good
**Contribution:** 2 fair
**Rating:** 5
**Confidence:** 4

**Summary:**

VideoDirectorGPT presents an innovative framework for consistent multi-scene video generation, utilizing the capabilities of GPT-4 for video content planning and scene description. The process begins with a single text prompt, which is expanded by the video planner LLM (GPT-4) into a comprehensive ‘video plan’, detailing scene descriptions, entity layouts, background settings, and consistency groupings. This information guides the Layout2Vid video generator, enabling explicit control over spatial layouts and maintaining temporal consistency across multiple scenes. VideoDirectorGPT demonstrates competitive performance against state-of-the-art models in single-scene text-to-video generation. The framework showcases potential for innovative applications, offers dynamic control features, and supports user interaction, setting a promising precedent for the integration of LLMs in long video generation and laying the groundwork for future advancements in the field.

**Strengths:**

1. Reasonable Pipeline Design: The framework adeptly utilizes GPT-4 for meticulous video content planning, optimally harnessing the extensive capabilities of this large language model to bring an innovative and groundbreaking approach to the realm of video generation.

2. Comprehensive Experimental Validation: The paper meticulously outlines a thorough and extensive experimental setup, ensuring a robust and all-encompassing evaluation of the framework’s performance and capabilities. It highlights a diverse range of scenarios and use cases, showcasing the framework’s exceptional versatility and its ability to seamlessly adapt to varying contexts.

3. Diverse Visual Illustrations: The paper and accompanying demo are enriched with a wide array of visual examples, vividly demonstrating the framework’s proficiency in generating multi-scene videos with varied themes and settings. Furthermore, the demo uniquely features support for user-provided images, thereby significantly enhancing user interaction and engagement with VideoDirectorGPT.

**Weaknesses:**

**Lack of Technical Contribution in Layout2Vid:** The Layout2Vid component of the VideoDirectorGPT framework, responsible for the actual video generation, appears to draw heavily from existing image generation work, particularly the Gilgen model which also operates based on layouts. There seems to be a noticeable lack of substantial technical differentiation or advancement in Layout2Vid, raising concerns about the novelty and contribution of this particular component to the field.

**Lack of Environment Consistency:** The VideoDirectorGPT framework, while innovative in its approach to multi-scene video generation, exhibits a notable lack of consistency in environmental elements across different scenes, as prominently seen in the "make caraway cakes" demo example. Although the object (the woman) maintains a consistent appearance throughout, the environment suffers from visible discontinuities, leading to a disjointed, montage-like effect in the resulting video. This issue seems to be a direct consequence of the framework’s heavy reliance on GPT-4 for generating scene descriptions, coupled with potential shortcomings in the Layout2Vid model's capability to translate these textual plans into visually cohesive sequences.

**Minor: Limited Developmental Space:** The director model in VideoDirectorGPT demonstrates a substantial dependency on GPT-4 for generating scene descriptions. This reliance not only makes the system vulnerable to any limitations and performance issues inherent in GPT-4, but it also raises questions about the developmental prospects of the director model. Given that GPT-4 is a pre-trained model with fixed capabilities, improvements in scene planning and description generation may be challenging to achieve without significant advancements in language model technology or a change in approach. Additionally, despite the innovative integration of GPT-4, the current state of long video generation still leaves much to be desired in terms of visual continuity and narrative coherence, indicating a need for further refinement and development.

**Questions:**

See the weakness above.

---

### Official Review · Reviewer_GLr8 · 2023-11-01

**Soundness:** 3 good
**Presentation:** 2 fair
**Contribution:** 2 fair
**Rating:** 5
**Confidence:** 5

**Summary:**

This paper elegantly deconstructs the process of video generation into two distinct phases: planning and execution. In the planning stage, GPT-4 takes on the director's role, meticulously crafting scene descriptions, arranging entities with their corresponding layouts, setting the backdrop for each scene, and ensuring consistency among the groupings of entities and backgrounds. Following this intricate planning process, the baton is passed to the video generation model, Layout2Vid. Leveraging the foundation laid by the pre-trained ModelScopeT2V, a Gated Self-Attention module has been fine-tuned, enabling the direct input of text, images, and layout as conditions for manifesting videos. Remarkably, it transcends the boundaries set forth by ModelScopeT2V, excelling in metrics such as accuracy and the spatial placement of generated entities.

**Strengths:**

1. The task of video generation has been dissected, enabling the application of LLM's knowledge into the realm of video creation.
2. The method of guiding video generation through layouts has been introduced to the task of video creation.
3. The training process of Layout2Vid is notably efficient.

**Weaknesses:**

1. As per the CLIPSIM indicator in Table 2 and Appendix F, even with the employment of the costly GPT-4 for video planning, it still lags behind Make-A-Video and VideoLDM, and is even outperformed by ModelScopeT2V. This suggests that there may be certain issues with Layout2Vid's model fine-tuning method.
2. As a T2V model, Layout2Vid's serious oversight lies in its failure to compare the FVD metrics with other models on UCF-101, a benchmark commonly used for T2V tasks.
3. While the paper claims its ability to generate long videos, it merely compares model capabilities with ModelScopeT2V within its own VideoDirectorGPT framework. This seems more like a comparison between Layout2Vid and ModelScopeT2V rather than a qualitative or quantitative comparison with other long video generation models such as Phenaki and NUWA-XL.
4. As a generation task, qualitative comparisons are crucial. However, the qualitative comparison in the paper only presents results between Layout2Vid and ModelScopeT2V, which is evidently inadequate.
5. The techniques employed on the Layout2Vid model are primarily based on ModelScopeT2V and GLIDEN, which is not novelty enough.

**Questions:**

1. The human evaluation results presented in Table 5 leave us wondering about the number of people involved in rating, as well as the fairness and reliability of the process?
2. Do you think that merely fine-tuning the Gated Self-Attention module might substantially diminish the original potent T2V generation capabilities of ModelScopeT2V?
3. The accesibility to GPT-4 is not always possible. If other open-sourced LLMs are used, would they still be able to generate video plans of the same high quality? Moreover, are the same prompts still effective for other LLMs?

---

> ### Author Response · Authors · 2023-11-18
>
> **W1 Did not outperform other baselines in CLIPSIM**:
>
>
> We would like to point out that our focus is on enabling the generation of consistent multi-scene videos, with LLM-based planning and layout guidance, while maintaining visual quality (FID/FVD) and video-text alignment (CLIPSIM). We demonstrate the usefulness of our method of layout control and multi-scene consistency in Tables 1 and 3. Moreover, as mentioned in Sec 5.1, while Make-A-Video and VideoLDM achieve better CLIPSIM than our VideoDirectorGPT, they are trained with larger video data or with higher resolution than our backbone (ModelScopeT2V). As our framework can flexibly use other video backbone models, we believe that our framework can achieve even better CLIPSim once we have access to those stronger backbone models.
>
>
> **W2 Missing UCF-101**:
>
>
> Since ModelScopeT2V doesn’t report their evaluation results on UCF-101 dataset in their technical report, we use their publicly released checkpoint and run evaluation on the same 2048 randomly sampled test prompts used in our VideoDirectorGPT evaluation. Compared with ModelScopeT2V, we achieve a significant improvement in FVD (ModelScopeT2V 1093 vs. Ours 748) and competitive performance in the Inception Score (ModelScopeT2V 19.49 vs. Ours 19.42).
>
>
>
>
>
>
> **W3 Comparison to multi-scene models**:
>
>
> Our model is not directly comparable with other long-video generation models based on the following two reasons. Firstly, the main focus of our model is multi-scene video generation with both layout and consistency control, which is not covered by any of the previous long video generation works. Secondly, most works on long video generation (e.g., NUWA-XL and Phenaki) are not open-sourced, therefore it’s hard for us to make a fair comparison. Specifically, NUWA-XL is trained on cartoon characters, and it’s hard for our VideoDirectorGPT to generate these cartoon characters without fine-tuning.
>
>
> **W4 The qualitative comparison only presents results between Layout2Vid and ModelScopeT2V**:
>
>
> The code and checkpoint of models such as NUWA, VideoLDM, MagicVideo, and Make-A-Video are not publicly accessible. Since ModelScopeT2V is the only open-source strong video backbone model that we could access, we built our Layout2Vid with ModelScopeT2V. We would be happy to experiment with our frameworks with other video backbone models when we can access them in the future.
>
>
>
>
> **Q1 Human Evaluation Setup**:
>
>
> As mentioned in Sec. 4, the human evaluation presented in the paper used three crowd-sourced workers from Amazon Mechanical Turk to evaluate each video. The workers were not given any information about which model generated each video (randomly shuffled) to ensure there was no bias towards either model. While we took the agreement of three workers per video, we had 19 unique workers to complete the human evaluation.
>
>
> We also extended the human evaluation to the agreement of 5 workers per video and a total of 28 unique workers. The results are shown below and VideoDirectorGPT still has a much higher human preference than ModelScopeT2V.
>
>
> |   | VideoDirectorGPT (Ours) | ModelScopeT2V | Tie |
> |---|:----:|:---:|:---:|
> | Quality  |     **52**    |   30  |  18 |
> | Text-Video Alignment |   **54**   |  22  |  24 |
> | Object Consistency   |    **62**    |   28  |  10 |
>
>
>
>
>
>
>
>
>
>
> **Q3 Open-sourced LLM Ablation**:
>
>
> We conducted an ablation study of using the open-sourced LLama2-13B-Chat model as well as GPT3.5-Turbo in the video planning stage on MSR-VTT dataset. Here we use LLama2 in a zero-shot evaluation setting without fine-tuning on layout-annotated videos. As we can see from the following table, LLama2 achieves worse scores on all three metrics (FID, FVD, and CLIPSIM), which shows that GPT-3.5/4 can generate more accurate layouts. Another point to notice is that GPT-3.5-turbo and GPT-4 achieve very close performance on MSR-VTT. One explanation for this is that MSR-VTT prompts don’t need strong layout control.
>
>
> |  | FID | FVD | CLIPSIM |
> |---|:----:|:-----:|:---------:|
> | GPT-4  |   12.22   | 550 | 0.2860 |
> | GPT-3.5-Turbo |  12.17    | 548 | 0.2862 |
> | LLama2-13B-Chat  | 13.47    | 573 | 0.2792 |
>
>
> To test this, we conduct another comparison of GPT-3.5-Turbo and GPT-4 on the ActionBench-Direction dataset we proposed in our paper. With $\alpha$=0.2, GPT-4 achieves around 10% higher accuracy than GPT-3.5-turbo in our movement direction metric. When we use LLM-dynamic-$\alpha$, such a gap increases to more than 20%. This indicates that (1) on datasets that need strong layout control, GPT-4 has a better performance compared with GPT-3.5-Turbo, and (2) GPT-4 is better at estimating the importance of spatial layouts from input prompts compared with GPT-3.5-Turbo.
>
>
> |  | Movement Direction Acc. ($\alpha$=0.2) | Movement Direction Acc. (LLM-dynamic-$\alpha$) |
> |----|:---:|:---:|
> | GPT-4   |   **59.8**  | **56.8** |
> | GPT-3.5-Turbo  |  49.0   | 35.2 |

---

### Official Review · Reviewer_tZEz · 2023-11-02

**Soundness:** 3 good
**Presentation:** 3 good
**Contribution:** 2 fair
**Rating:** 3
**Confidence:** 4

**Summary:**

The paper presents VideoDirectGPT, a novel framework for generating consistent multi-scene videos by leveraging large language models (LLMs) for video content planning and grounded video generation. It expands a text prompt into a 'video plan' using an LLM, enabling explicit control over spatial layouts and ensuring temporal consistency across scenes, achieving improved video quality and movement control.

**Strengths:**

In a word, the paper proposes a straightforward solution for video generation by leveraging GPT planning capability and pretrained text-to-video model.

**Weaknesses:**

1. How does the unclip prior affect the video quality and text image alignment?

2. What if representations are not shared?

3. Except for using GPT an the planner, the novelty is quite limited.  In particular, compared with both GLIGEN and ModelScopeT2V, the only contribution is unclip prior?

4. Have the authors tried only finetune gated self-attn layer? What does the performance look like?

5. Does the baseline ModelScopeT2V use the same dataset as the papers uses for fine-tuning?

6. Comparing ModelScopeT2V and the proposed method on the actionbench-direction prompts, the object is actually not moving. Quality is not good as the baseline. And the advantage of text guidance actually comes from layout control, making the method contribution rather limited.

**Questions:**

Please see above.

---

> ### Author Response · Authors · 2023-11-18
>
> **W1 How does unclip prior affect video quality and image-text alignment?**:
>
>
> We’d like to first briefly remind the reason for using unCLIP prior. During training, we can obtain image embedding of entity representation by bounding box crop of the entities in the target image. During inference, since we don’t have access to the target image to crop from, we employ the unCLIP prior to obtain CLIP image embeddings from the entity text descriptions (see appendix C2).
>
>
> In Figure 6, we show that the image embedding from unCLIP prior can effectively represent its original text description (‘white cat’). In our single- and multi-scene video generation results (Tables 2 and 3), we have shown qualitatively that our VideoDirectorGPT achieves comparable or better scores in video quality (FID/FVD) and video-text alignment (CLIPSIM).
>
>
>
>
> **W2 What if the representations are not shared**:
>
>
> We believe this is a misunderstanding of our method. If an entity (for example, a cat) should maintain visual consistency across two different scenes, then we use the same CLIP image+text embeddings to represent the cat when constructing grounding tokens. Instead, if the cats in scene1 and scene2 are different, then we use different CLIP image+text embeddings to represent them respectively.
>
>
>
>
>
>
>
>
> **W4 Only finetuning gated-self attention layer**:
>
>
> Thanks for pointing this out.
> In the Guided 2D Attention module, only the Gated Self-Attention layers are trainable, while all other layers (self-attention and cross-attention layers) are kept frozen.
>
>
>
>
> **W5 Does the baseline ModelScopeT2V use the same dataset as the papers uses for fine-tuning?**:
>
>
> We would like to remind the reviewer that all the experiments on single- and multi-scene video generation are based on zero-shot. Both our VideoDirectorGPT and the benchmark models (including ModelScopeT2V) are not specifically fine-tuned for the benchmark datasets.
>
>
> In addition, our video generation module, Layout2Vid, is built upon the ModelScopeT2V backbone with additional Gated Self-Attention layers. Only these gated self-attention layers are trained, while all parameters in the original ModelScopeT2V backbone are kept frozen. The dataset we used for training is the same as the one used in GLIGEN, which contains only image-level annotations without any video-level data (see Sec. 3.2) .

---

### Official Review · Reviewer_pdWu · 2023-11-09

**Soundness:** 2 fair
**Presentation:** 2 fair
**Contribution:** 3 good
**Rating:** 3
**Confidence:** 4

**Summary:**

This paper proposes a two-stage text-to-video generation framework, consisting of video content planning and grounded multi-scene video generation. The first module employs a large language model (LLM), such as GPT-4, to generate a video plan. The second module, trained with image-level layout annotations, generates a consistent multi-scene video given the video plan. The authors conduct various evaluations to demonstrate the effectiveness of their work.

**Strengths:**

1. The proposed framework achieves high efficiency by not requiring video training data and maintaining good results with 87% of total parameters fixed.
2. The authors develop several novel evaluation methods that provide solid comparisons between the proposed framework and previous works.
3. The framework uses both high-level and low-level conditioning to enable fine-grained control over generated videos.
4. Intuitively and effectively, the framework uses shared features for the same subject across different scenes to ensure multi-scene consistency.

**Weaknesses:**

### A. Main paper
1. This paper is somewhat too abstract throughout. The introduction is adequate, but I would expect to see more technical content in the following sections. For example, the loss function for the proposed image fine-tuning is not provided. Additionally, there is little evidence to support the correctness of the proposed methods beyond empirical results.
2. The introduction to the datasets is limited. Some datasets provide fine-grained descriptions for each scene, while others only provide a single sentence for an entire video. Furthermore, the authors customize the Pororo-SV dataset by replacing character names with pronouns, but they do not justify this procedure. The lack of a clear explanation of the datasets makes it difficult to understand the task, such as the inputs and outputs for training and testing, and whether a large language model (LLM) is used.
3. An ablation study should be conducted to demonstrate the effectiveness of the LLM. Additionally, if the LLM was used to refine prompts, these prompts should also be given to ModelScopeT2V to enable a fair comparison and provide readers with better insights.
4. Since the consistency should be maintained regardless of the temporal distance between scenes, the authors should consider using the variance of CLIP features of all scenes instead of the average of similarities across adjacent scene pairs.
5. The human evaluation does not have enough participants to provide reliable results.

### B. Qualitative results
1. The objects not exactly follow the bounding boxes.
2. The "pushing object" video examples appear to show camera movement rather than object movement.

**Questions:**

1. How do you replace the original animation characters in Pororo-SV with real-world entities? Do the edited videos look natural enough? Why do you use pronouns to replace character names, and wouldn't this make it difficult for ModelScopeT2V to guess the correct content, leading to an unfair comparison?
2. What is the exact loss function used for finetuning?
3. Why are some numbers not available in the results, such as FVD and FID for Coref-SV and Consistency for HiREST?

---

> ### Author Response · Authors · 2023-11-18
>
> **W.A-2 / Q.1 Pororo-SV dataset details**:
>
>
> As mentioned in the main paper section 4 and appendix D.3, we modified the text descriptions of the Pororo-SV to create Coref-SV. We replace character names with basic animals and then duplicate instances of character names with a co-reference. For example, if the description is “Pororo got out of bed. Pororo went to the kitchen to get a snack.” then the Coref-SV version of the prompt would be “Dog got out of bed. He went to the kitchen to get a snack.” We do not use an LLM to complete this. We do not modify the videos of Pororo-SV either. The purpose of doing this is to measure whether or not a model can maintain the visual consistency of a character across scenes without explicitly being told that they are the same object. This exemplifies the benefit of using an LLM planner to create “video plans” as the LLM is able to understand this coreference and map it to the same consistency grouping across scenes, whereas existing models like ModelscopeT2V would get confused.
>
>
>
>
> **W.A-3 Ablation of using LLM-expanded prompts in ModelScopeT2V**:
>
>
> Following your suggestion, we experiment with generating videos with ModelScopeT2V using the same scene descriptions from our video plans (produced by GPT-4), on the HiREST dataset. As we can see from the following table, ModelScopeT2V with LLM-expanded prompts (2nd row) achieves better performance than the original ModelsScopeT2V (1st row), showing the effectiveness of our video plans. However, it still performs worse than our VideoDirectorGPT (3rd row), also showing the effectiveness of our Layout2Vid.
>
>
>
>
> | HiREST  | FVD |  FID   |
> |-----|:---:|:-----:|
> | ModelScopeT2V | 1322 | 23.79 |
> | ModelScopeT2V (with prompts from our video plan) |  918 | 18.96  |
> | VideoDirectorGPT (Ours)  | **733** | **18.54** |
>
>
>
>
>
>
> **W.A-4 CLIP Variance based consistency metric**:
>
>
> As per your suggestion, we compute variance of the CLIP features across all scenes. Results are shown in the table below (lower is better). As you can see, our model still outperforms ModelscopeT2V, even when we give it the advantage of the GT co-references.
>
>
> |  | CLIP image embedding variance * 1000 (Lower is better) |
> |----|:---:|
> | ModelScopeT2V  |    0.59  |
> | ModelScopeT2V (with GT co-reference; oracle) |  0.48  |
> | VideoDirectorGPT (Ours) |  **0.46**   |
>
>
>
>
>
>
> **W.A.5 Not enough participants for human evaluation**:
>
>
> As mentioned in Sec. 4, the human evaluation presented in the paper used three crowd-sourced workers from Amazon Mechanical Turk to evaluate each video. The workers were not given any information about which model generated each video (randomly shuffled) to ensure there was no bias towards either model. While we took the agreement of three workers per video, we had 19 unique workers to complete the human evaluation.
>
>
> To address the concern that there were not enough participants, we extended the human evaluation to the agreement of 5 workers per video and a total of 28 unique workers. The results are shown below and VideoDirectorGPT still has a much higher human preference than ModelScopeT2V.
>
>
> |   | VideoDirectorGPT (Ours) | ModelScopeT2V | Tie |
> |---|:---:|:---:|:---:|
> | Quality |          **52**         | 30 | 18 |
> | Text-Video Alignment |  **54** | 22 | 24 |
> | Object Consistency | **62** | 28 | 10 |
>
>
>
>
>
>
>
>
> **W.B-1 Objects not exactly following bounding boxes**:
>
>
> In our video generation module Layout2Vid, the strength of layout control is determined by the hyper-parameter $\alpha$, which represents the number of denoising steps we activate the Gated Self-Attention layers. High $\alpha$ value yields stronger layout control, with the cost of reducing visual quality, which is also observed in other T2I generation works [1, 2, 3]. We found that an $\alpha$ value between 0.1-0.3 usually gives the best performance, and we set it as 0.1 (which is equivalent to 5 denoising steps) as the default value (as shown in our ablation study of $alpha$ hyper-parameter in Sec. 5.3 of our paper). Therefore, the bounding boxes in our frameworks are more about providing layouts and motion guidance, rather than strict restriction of object locations.
>
>
> [1] Li et al., GLIGEN: Open-Set Grounded Text-to-Image Generation (CVPR 2023)
>
> [2] Cho et al., Visual Programming for Text-to-Image Generation and Evaluation (NeurIPS 2023)
>
> [3] Lian et al., LLM-grounded Diffusion: Enhancing Prompt Understanding of Text-to-Image Diffusion Models with Large Language Models

---

### Author Response · Authors · 2023-11-18
**General response**

We thank the reviewers for their feedback and for recognizing our strengths:

- Highly efficient training: without video annotation / 87% of parameters are frozen (pdWu,GLr8)
- Novel evaluations with solid comparisons between models (pdWu)
- Enables fine-grained control over generated videos (pdWu)
- Ensures multi-scene consistency intuitively / effectively (pdWu)
- Enables the combination of LLM’s knowledge and pretrained text-to-video generation models (tZEz,GLr8)
- Intuitive / reasonable pipeline design (pdWu,dMSD)
- Innovative / groundbreaking approach (dMSD)
- Comprehensive experimental validation, covering diverse use cases (dMSD)
- Diverse visual illustrations (dMSD)
- Supports user-provided images, significantly enhancing user interaction (dMSD)

**With our several method and evaluation contributions (for the new task of consistent multi-scene long video generation) and with the minor questions/ablations asked, we believe that our submission received unwarranted low scores (hence we are withdrawing our paper). We still add responses to the novelty question below (and the requested minor experiments/ablation results in individual responses).**


Regarding the novelty of our method, we would like to clarify the following:

1\. **The primary contribution** of VideoDirectorGPT is providing multi-scene video generation by leveraging the LLM knowledge for planning and ensuring consistency groups across scenes.

2(a). **Our technical novelty of video planner**. We present the first work of teaching an LLM to generate a complex multi-component video plan (scene descriptions, entity descriptions and layouts, and consistency groupings of entities/backgrounds) for multi-scene video generation. Also, many recent papers exploring the planning capabilities of LLM have been well-received in communities without introducing new architectures [1,2,3].

2(b). **Our technical novelty of Layout2Vid.** No previous work in text-to-video generation has proposed techniques that can achieve spatial control, temporal consistency, and also training efficiency (without video annotations / parameter-efficient training) at the same time. The original unCLIP and GLIGEN models are introduced for text-to-image generation and layout-guidance in image generation, and it is not trivial to adapt/combine these modules for improving video generation.

3\. **We also have made big contributions in evaluation metrics and extensive experiments.** In addition to the methodological contributions, we also provide comprehensive qualitative/quantitative experiments including novel evaluation metrics (movement direction accuracy / entity consistency) /new datasets (ActionBench-Direction/Coref-SV) for video generation, as pointed out by other reviewers.


[1] Gupta and Kembhavi, Visual Programming: Compositional Visual Reasoning Without Training (CVPR 2023 Best paper)

[2] Surís et al., ViperGPT: Visual Inference via Python Execution for Reasoning (ICCV 2023)

[3] Cho et al., Visual Programming for Text-to-Image Generation and Evaluation (NeurIPS 2023)